TOPICAL REVIEW

# The role of immune responses and microbiota in adipose tissue homeostasis

Shahar Azar and Barbara Rehermann 

*Immunology Section, Liver Diseases Branch, National Institute of Diabetes, Digestive and Kidney Diseases, National Institutes of Health, DHHS, Bethesda, Maryland, USA*

Handling Editors: Kim Barrett & Stephen Keely

The peer review history is available in the Supporting Information section of this article (https://doi.org/10.1113/JP286422#support-information-section).

**Abstract figure legend** Gut microbiota regulate white and brown adipose tissue homeostasis via modulation of gut hormone secretion, activation of pathogen-associated pattern receptors and production of bioactive molecules and metabolites such as short-chain fatty acids, endocannabinoids, amino acid derivates and bile acids, all of which affect adipocytes, adipose tissue homeostasis and host metabolism (upper and lower arrows). Evidence for an evolutionarily conserved microbiota–immune cell–adipose tissue axis (middle arrow) is proposed and discussed in this review.

**Abstract** White and brown adipose tissue form a metabolic organ that plays a crucial role in regulating body energy homeostasis. Adipose tissue is richly vascularized and innervated to respond to a variety of environmental signals. Adipose tissue also contains diverse populations of innate and adaptive immune cells. These immune cells contribute to the regulation of adipose tissue function, and adipocytes in turn signal to immune cells in response to metabolic

**Shahar Azar** is a postdoctoral research fellow in the Immunology Section, Liver Diseases Branch, NIDDK, at the National Institutes of Health (NIH) in Bethesda, Maryland, USA. He received his PhD degree from the Institute for Drug Research, School of Pharmacy, Faculty of Medicine, the Hebrew University of Jerusalem, Israel. During his graduate studies he investigated the contribution of the endocannabinoid system to metabolic dysfunction-associated fatty liver disease in obesity. Currently he studies the effect of the microbiota on the immune system and how both affect host metabolism. **Barbara Rehermann** is a physician scientist and chief of the Immunology Section, Liver Diseases Branch, NIDDK, NIH. She received an MD degree and the Venia Legendi for Immunology from Medizinische Hochschule, Hannover, Germany, followed by clinical training in the Department of Gastroenterology, Hepatology and Endocrinology and postdoctoral research at the Scripps Research Institute, La Jolla, California. Her laboratory at the NIH performs translational immunology studies with well-characterized patient cohorts, and preclinical mouse models with natural microbiota to evaluate the regulation of hepatic and systemic immune responses and metabolism.

The Journal of Physiology

and environmental triggers. The gut microbiota have recently emerged as an additional factor that affects adipose tissue homeostasis. This can occur either directly via metabolites and bacterial products or indirectly via its effects on immune cells. Natural, co-evolved microbiota, if encountered in early postnatal life, have been shown to confer protection against obesity in later periods of life. The complexity of these factors and interactions warrants further investigation and may ultimately provide opportunities for therapeutic interventions that prevent obesity and metabolic disease.

(Received 30 June 2025; accepted after revision 14 October 2025; first published online 27 November 2025)

**Corresponding author** B. Rehermann: Immunology Section, Liver Diseases Branch, National Institute of Diabetes, Digestive and Kidney Diseases, National Institutes of Health, DHHS, Bethesda, MD 20892, USA.    Email: rehermann@nih.gov

## Introduction

Mammals have three types of adipose tissue, namely white, brown and beige adipose tissue. White adipose tissue (WAT) serves as the body's energy reservoir, accumulating excess energy in the form of triglycerides in times of energy surplus and supplying lipids as energy source to other tissues in times of energy deprivation. Brown adipose tissue (BAT) dispenses energy in the form of heat in a process termed 'non-shivering thermogenesis', thereby contributing to both thermoregulation and energy expenditure. Beige adipocytes are located within WAT but are functionally more like brown adipocytes with the capacity to be thermogenically active. Adipose tissue also secretes factors that exert systemic effects on energy balance and metabolism. Thus, functionally intact adipose tissue is important for both metabolic homeostasis and energy balance.

Immune responses play a major role in the maintenance of adipose tissue function. Both white and brown adipocytes signal to immune cells, and innate and adaptive immune cells affect adipocytes (Trim & Lynch, 2022). Alterations in immune responses with downstream effects on adipocytes have been observed in obesity and secondary to changes in the gut microbiota. As microbiota and commensals play an essential role in the induction and tuning of tissue-resident immune responses (Belkaid & Harrison, 2017) and have co-evolved with mammals in a symbiotic relationship to support their metabolic health (Chilloux et al., 2016), it is intuitive to postulate the existence of a microbiota–immune cell–adipose tissue nexus. This review provides a primer on adipose tissue physiology and function and on the role of immune responses in BAT and WAT homeostasis *versus* obesity-induced inflammation. It concludes discussing a microbiota–immune cell–adipose tissue nexus and identifies research questions and challenges for future work.

**Adipose tissue location and function.** WAT, beige adipose tissue and BAT differ in their anatomical location, function and morphology. WAT is the most abundant type of adipose tissue (Cypess, 2022; Zwick et al., 2018) and divided into subcutaneous and visceral WAT. In humans subcutaneous WAT represents 80% of the total fat mass and is located in the abdominal and gluteofemoral areas (Zwick et al., 2018). Visceral WAT accounts for the remaining 5%–20% of the fat mass and is located in the mesenteric, omental, retroperitoneal, perirenal, gonadal and pericardial regions. The major visceral depot in humans is the omentum, which drains into the portal circulation and the liver (Fig. 1*A*).

In rodents, subcutaneous WAT is located in the axillary and inguinal regions, and visceral WAT is located in the mesenteric, perirenal, retroperitoneal and perigonadal regions. The major visceral WAT depot in rodents is the perigonadal WAT. It drains into the systemic circulation, bypassing the liver (Fig. 1*B*) (Sakers et al., 2022). WAT's primary role is to store excess energy in the form of triglycerides and to release lipids in response to the body's energy demands (Rosen & Spiegelman, 2014). WAT additionally serves as an active endocrine organ, which regulates whole-body metabolism, energy homeostasis, insulin sensitivity and inflammatory processes (Cypess, 2022; Funcke & Scherer, 2019; Hotamisligil et al., 1993; Sakers et al., 2022).

BAT is less abundant than WAT. It develops before birth (Cohen & Kajimura, 2021; Cypess, 2022; Ikeda et al., 2018) and is most prevalent in infants (Cypess, 2023; Gilsanz et al., 2013). However it still accounts for $\sim$0.2%–3% of the total adipose tissue mass and 0.1%–0.5% of the total body weight in adults (Cypess, 2022, 2023; Sacks & Symonds, 2013), where it is located in the supraclavicular, interscapular, neck, paraspinal and abdominal areas (Cypess et al., 2009; Leitner et al., 2017; Saito et al.,

2009; van Marken Lichtenbelt et al., 2009; Virtanen et al., 2009; Zhang et al., 2018). Rodents are the most common model for studying BAT physiology. Because they are three orders of magnitudes smaller than humans and have a higher surface-to-volume ratio than humans, they are more susceptible to hypothermia. BAT is essential to maintain their body core temperature (Cypess, 2022, 2023) and represents 2%–5% of their total body weight.

In humans and rodents, BAT is extensively vascularized, highly innervated and equipped with a high density of iron-rich mitochondria (Cohen & Kajimura, 2021). These features support its primary role in responding to noradrenaline (norepinephrine) from sympathetic neurons with non-shivering thermogenesis (Rosen & Spiegelman, 2014). The stimulation of $\beta$3-adrenergic receptors and the downstream cAMP-PKA pathway in brown adipocytes results in lipolysis and increased expression of thermogenic genes, including uncoupling protein 1 (UCP1) (Morrison & Madden, 2014). UCP1 is a mitochondrial protein that dissociates nutrient oxidation from ATP synthesis by dispersing the inner mitochondrial proton gradient, thereby generating heat. In addition to UCP1-dependent thermogenesis, several UCP1-independent mechanisms of thermogenesis are known. These include the futile creatine cycle, which involves the synthesis and hydrolysis of phosphocreatine (Sun et al., 2021), the futile calcium cycle (Ikeda & Yamada, 2022) and the futile lipid cycle (Oeckl et al., 2022).

Beige adipocytes develop postnatally in WAT by a process termed 'beiging' or 'browning' in response to different stimuli that include exposure to cold, stimulation with $\beta$3-adrenergic receptor agonists, exercise and injury (Cohen & Kajimura, 2021; Ikeda et al., 2018). Beige adipocytes are considered a distinct cell type based on a unique gene signature (Ikeda et al., 2018; Wu et al., 2012). They are characterized by numerous multilocular lipid droplets, high mitochondrial density and increased UCP-1 expression (Cohen & Kajimura, 2021), and thus, are involved in non-shivering thermogenesis. Noradrenaline-induced release of fibroblast growth factor-21 (FGF21) induces browning of WAT and thermogenic activity and CCL11-mediated recruitment of eosinophils. The latter release interleukin-4 (IL-4) and thereby stimulate the proliferation of beige adipocyte precursors (Huang et al., 2017).

Within all adipose tissue depots there is a high degree of heterogeneity as regards to adipose stem cells, progenitor cells and mature adipocytes (Emont et al., 2022; Sun et al., 2020). This heterogeneity has been observed in both humans and mice (Emont et al., 2022) and is functionally relevant as regards differential activation of metabolic pathways from triglyceride biosynthesis to insulin signalling.

**Adipose tissue and its role in metabolic homeostasis.** Like no other organ WAT can expand and contract. WAT is responsible for 5% of the insulin-mediated glucose uptake in lean adults and for 20% in obese adults (Cypess, 2022; Sakers et al., 2022). In times of energy surplus white adipocytes absorb lipids from the circulation or convert other nutrients to lipids via *de novo* lipogenesis. WAT enlarges by increasing both the number of adipocytes (hyperplasia, which typically improves fat storage and metabolic parameters) and the size of individual adipocytes (hypertrophy, which is typically associated with inflammation, hypoxia and fibrosis). Adipocyte hyperplasia is limited under normal conditions relative to hypertrophy.

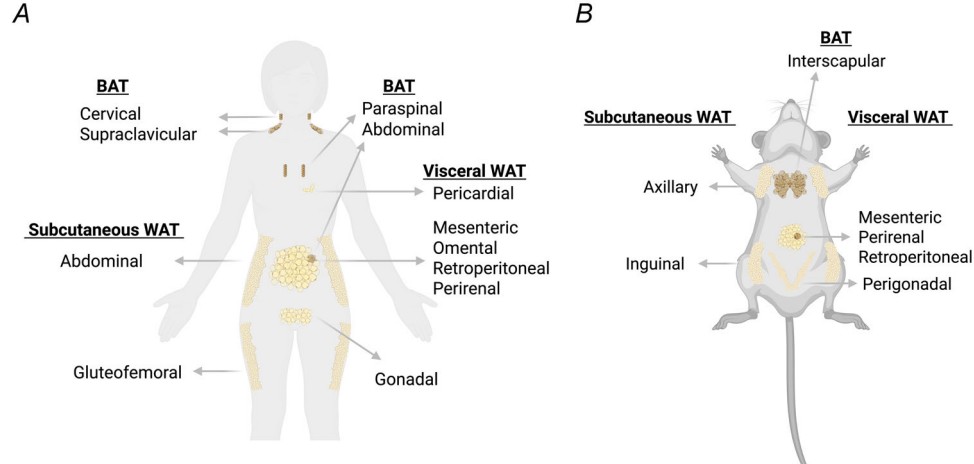

**Figure 1. Distribution of brown adipose tissue (BAT) and white adipose tissue (WAT)**
The distribution of BAT and WAT is shown in *A*, humans and in *B*, mice, a common model for basic and preclinical studies.

In times of fasting or high energy demand WAT releases stored triglycerides in the form of free fatty acids. This process of lipolysis is stimulated by catecholamines from sympathetic neurons and inhibited by insulin (Maniyadath et al., 2023; Sakers et al., 2022). Sympathetic stimulation after cold exposure also drives – via activation of $\beta$3-adrenergic receptors – browning of WAT and thermogenesis of BAT. Therefore, $\beta$3-receptor agonists increase BAT metabolic activity (O'Mara et al., 2020). This process is associated with increased clearance of fatty acids, triglycerides and glucose from the blood in both lean and obese mice (Bartelt et al., 2011; Labbe et al., 2016; Shin et al., 2022; Worthmann et al., 2019) and in humans (Blondin et al., 2014; Chondronikola et al., 2016; Maliszewska & Kretowski, 2021; Ouellet et al., 2012; Saito et al., 2009).

In addition to their roles in utilizing nutrients to store and dissipate energy, WAT and BAT secrete factors, such as leptin and adiponectin, that affect systemic metabolism and energy balance. Leptin is secreted from white adipocytes in approximate proportion to their triglyceride content. It signals satiety in the hypothalamus and other brain regions that regulate energy balance (Funcke & Scherer, 2019; Pan & Myers, 2018). Adiponectin acts on cells of the liver, kidney, pancreas and muscles, and on immune cells among others, and stimulates beneficial metabolic processes such as sensitization to insulin and anti-inflammatory effects. Adiponectin levels decrease in the obese state, whereas leptin levels do not (reviewed in Funcke & Scherer, 2019; Hui et al., 2015; Iacobellis et al., 2013; Straub & Scherer, 2019; Wang et al., 2015). More evidence on the endocrine role of BAT is summarized and reviewed elsewhere (Yang & Stanford, 2022).

**Immune cell landscape in adipose tissue.** Although mature adipocytes account for 90% of the adipose tissue volume, they constitute less than 50% of its cells. In addition to preadipocytes the remaining cells include immune cells, fibroblasts, vascular endothelial cells and neurons, collectively named the stromal vascular fraction (Corvera, 2021; Emont et al., 2022; Maniyadath et al., 2023). Although WAT has a higher immune cell content than BAT (Fitzgibbons et al., 2011), all innate and adaptive immune cell populations are present in both tissues, and their relative abundance changes in response to nutritional and environmental signals (Hildreth et al., 2021). The following section describes key aspects of the immune landscape in adipose tissue homeostasis and inflammation.

*Pleiotropic roles of adipose tissue macrophages.* Macrophages constitute the most abundant immune cell population in adipose tissue (Emont et al., 2022). They are derived from two sources with distinct functions. One source is yolk-sac myeloid cells that seed into WAT and BAT during organogenesis, where they remain as self-renewing, tissue-resident $CD64^+MerTK^+$ macrophages (Gallerand et al., 2021). The other source is monocytes that are derived from haematopoietic stem cells in the bone marrow. These are continuously recruited throughout life in a CCR2 (C-C chemokine type 2 receptor)-dependent manner and are $CD64^+MerTK^-$ (Gallerand et al., 2021; Gautier et al., 2012). Because the macrophage pool in BAT requires constant replenishment, depletion of monocytes prevents tissue expansion (Gallerand et al., 2021).

Tissue-resident macrophages play an important role in WAT and BAT homeostasis, and several specific examples of specialized macrophage function in adipose tissue are given in the following section. In WAT, tissue-resident macrophages regulate synthesis and storage of lipids (Fig. 2*A*), and high-fat-diet induces platelet-derived growth factor cc (PDGFcc) production specifically in $Tim4^+$-resident macrophages. The released PDGFcc downregulates suppressors of lipid synthesis and storage in white adipocytes, thereby increasing lipid synthesis and storage. Conversely, blocking of PDGFcc prevents lipid storage in mice on a high-fat diet (Cox et al., 2021; Valdearcos et al., 2017). In BAT, the number of tissue-resident macrophages and monocytes increases after cold exposure (Burl et al., 2022), and these are thought to contribute to thermogenesis by removing damaged mitochondria (Rosina et al., 2022).

Tissue-resident macrophages also regulate WAT and BAT innervation (Figs 2*B* and 3). This was first suggested in a study by Pirzgalska and colleagues, who identified sympathetic neuron-associated macrophages (SAMs) in WAT. These SAMs exhibit an altered shape and increased expression of transcripts of genes that are involved in synaptic signalling, cell–cell adhesion and neuron development (Pirzgalska et al., 2017). They reduce the sympathetic tone in the adipose tissue by clearing noradrenaline, which they take up via the noradrenaline transporter solute carrier family 6 member 2 (SLC6A2) and subsequently degrade. Accordingly, mice with SLC6A-deficient SAMs respond to cold challenge with greater sympathetic activation and higher thermogenic activity, and transplantation of SLC6A2Sdeficient SAMs into genetically obese mice prevents adipocyte hypertrophy (Pirzgalska et al., 2017). In BAT $Lyve1^{low}CX3CR1^{high}MHC-II^{high}$ macrophages were described near nerve fibres (Chakarov et al., 2019; Wolf et al., 2017) and shown to regulate sympathetic innervation. Mice with a macrophage-specific deletion of the gene encoding transcription factor methyl-CpGbinding protein 2 (*Mecp2*) exhibit impaired innervation and thermogenic activity of BAT and become obese (Wolf et al., 2017). This is attributed to *Mecp2*-deficient macrophages overexpressing Plexin4, a molecule that contributes to

the repulsion of semaphorin 6A-expressing sympathetic neurons. Thus, *Mecp2*-deficient, Plexin4-expressing macrophages can repress sympathetic BAT innervation (Wolf et al., 2017).

Macrophages also play a role in extracellular matrix remodelling (Gallerand et al., 2021), a necessary process that enables healthy adipose tissue expansion. In obesity this accumulation of extracellular matrix components can become excessive. In this condition, thickened extracellular matrix contributes to hypoxia and adipocyte death, which along with fibrosis contributes to adipose tissue inflammation (reviewed in Sun et al., 2023).

Single-cell RNA sequencing results support the notion of a spectrum of macrophage subpopulations in adipose tissue that range from athe classically activated M1 state macrophages (Kang & Lee, 2024; Maniyadath et al., 2023), which accumulate in obese adipose tissue (Kratz

et al., 2014), to the more metabolically active M2 state macrophages (also called alternatively activated, anti-inflammatory macrophages. The anti-inflammatory M2 phenotype dominates in lean adipose tissue and is thought to result from the expression of the transcription factor peroxisome proliferator-activated receptor gamma (PPAR$\gamma$) (Odegaard et al., 2007). M2 macrophages also exhibit increased expression of the mannose receptor CD206 (*Mrc1*), the C-type lectin CD301 (*Clec10a*), and secrete arginase1, an IL-1 receptor antagonist, and transforming growth factor $\beta$ (TGF-$\beta$), both of which are anti-inflammatory and contribute to tissue repair. M2 macrophages promote beige adipogenesis via multiple mechanisms, including secretion of PPAR$\gamma$ ligands that bind to beige precursor cells (Chung et al., 2017). Secretion of catecholamines was also reported (Nguyen et al., 2011) but refuted in later studies (Fischer et al.,

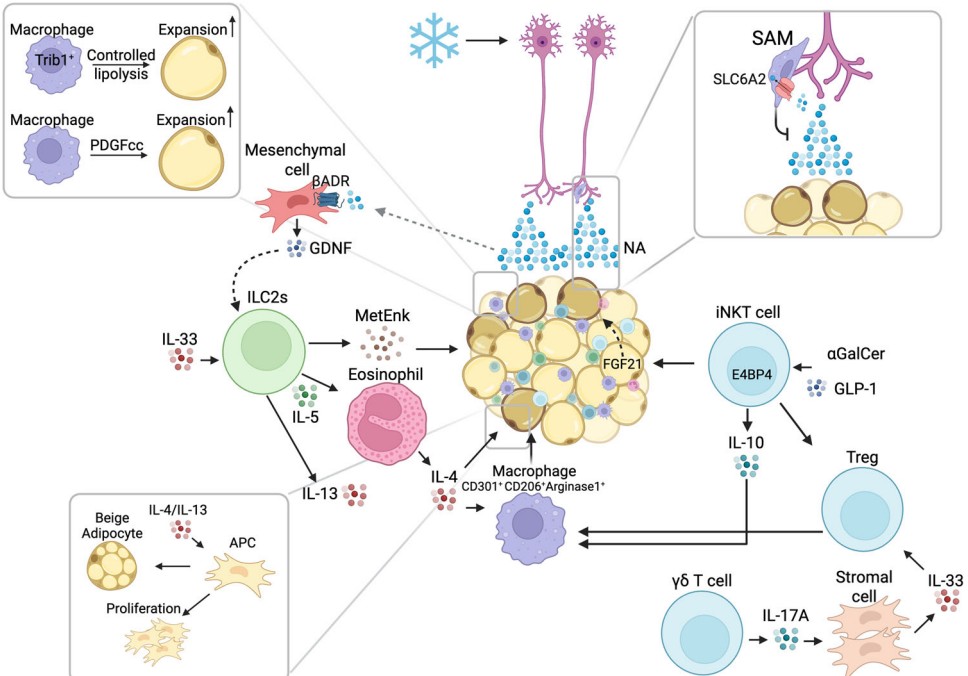

**Figure 2. Immune cells regulate white adipose tissue (WAT) expansion and beiging**
Adipose tissue-resident macrophages secrete PDGFcc which regulates lipid storage, and Trib1[+] macrophages regulate lipolysis. Both mechanisms support adipose tissue expansion. Sympathetic neuron-associated macrophages (SAMs) reduce the sympathetic tone of the tissue by clearing noradrenalin, thereby inhibiting beiging. Sympathetic signalling stimulates $\beta$-adrenergic receptors in mesenchymal cells, which leads to GDNF-mediated activation of ILC2s. In addition $\gamma\delta$ T cells induce stromal cells to produce IL-33, which in turn induces the proliferation of anti-inflammatory T$_{regs}$ (regulatory CD4[+] T cells) and activates ILC2s. Activated ILC2s produce methionine-enkephalin peptide, resulting in beiging of WAT. ILC2-derived IL-4, IL-5, IL-10 and IL-13 stimulate IL-4 production by eosinophils, catecholamine secretion by alternatively activated M2 macrophages and proliferation of adipocyte progenitors and their commitment to the beige linage. iNKT (invariant NKT) cells, activated by GLP-1 (glucagon-like peptide-1) or lipid antigens, also stimulate beiging via FGF21 (fibroblast growth factor-21) production and production of type 2 cytokines. Other iNKT cells (not depicted) can exert cytotoxic function either against adipocytes or (via IL-10-mediated licensing of NK cells) against macrophages (see text for detail). ADR, adrenalin receptor; APC, adipocyte progenitor cell; GDNF, glial cell-derived neurotrophic factor; NA, noradrenaline; PDGFcc, platelet-derived growth factor cc; SAM, sympathetic neurons associated macrophages; SLC6A2, solute carrier family 6 member 2, a norepinephrine transporter; Trib1, tribbles pseudokinase 1, an adapter protein involved in protein degradation. The ice crystal represents a cold stimulus.

2017). M2 macrophages can prevent inflammation by removing remnants of dead adipocytes that are produced as a result of hypoxia and mechanical stress in hypertrophic adipose tissue (Lee et al., 2013). M2 macrophages also contribute to the control of lipolysis and deficiency in tribbles pseudokinase 1 (Trib1), a protein that is essential for M2 macrophage differentiation, is associated with increased lipolysis and hypertriglyceridaemia in mice on a high-fat diet (Satoh et al., 2013).

The macrophage population in adipose tissue is replenished by recruitment and differentiation of monocytes from the bone marrow in response to specific adipose tissue cues. BAT, for example, secretes the chemokine C-X-C motif chemokine ligand-14 (CXCL14) in response to thermogenic activation, which results in increased recruitment of M2 macrophagesin support of BAT thermogenesis, WAT beiging and improvement in metabolic parameters (Cereijo et al., 2018). IL-6, released by both brown adipocytes and immune cells, promotes alternative activation of these M2 macrophages, thermogenic activity of BAT and beiging of WAT. Additionally it exerts systemic metabolic effects, including sensitization to insulin (reviewed in Bienertova-Vasku et al., 2018).

In obese tissue, bone marrow-derived monocytes expand into a macrophage population with an M1-like phenotype (Amano et al., 2014; Emont et al., 2022; Weisberg et al., 2003). This process is driven by proinflammatory cytokines and mediators that are released as a result of adipose tissue hypoxia and mechanical stress or as a result of bacterial antigens and free fatty acids activating pattern recognition receptors (Reilly & Saltiel, 2017). Adoptive transfer of proinflammatory M1 macrophages into lean mice induces a WAT gene expression profile that is similar to that observed in obesity (Hill et al., 2018) and promotes inflammation and metabolic syndrome (Cox & Geissmann, 2020; Li et al., 2023). M1 macrophages are a main source of TNF-$\alpha$, IL-1$\beta$ and IL-10, which repress thermogenic genes in adipocytes and promote sterile inflammation (Goto et al., 2016). IL-10 levels in both WAT and the systemic circulation are positively correlated with body mass index and adverse metabolic phenotypes in humans (Acosta et al., 2019; Esposito et al., 2003). Conversely global deletion of IL-10 or adipocyte-specific deletion of the IL-10 receptor-$\alpha$ increases the expression of thermogenic genes and thermogenic adipocytes in

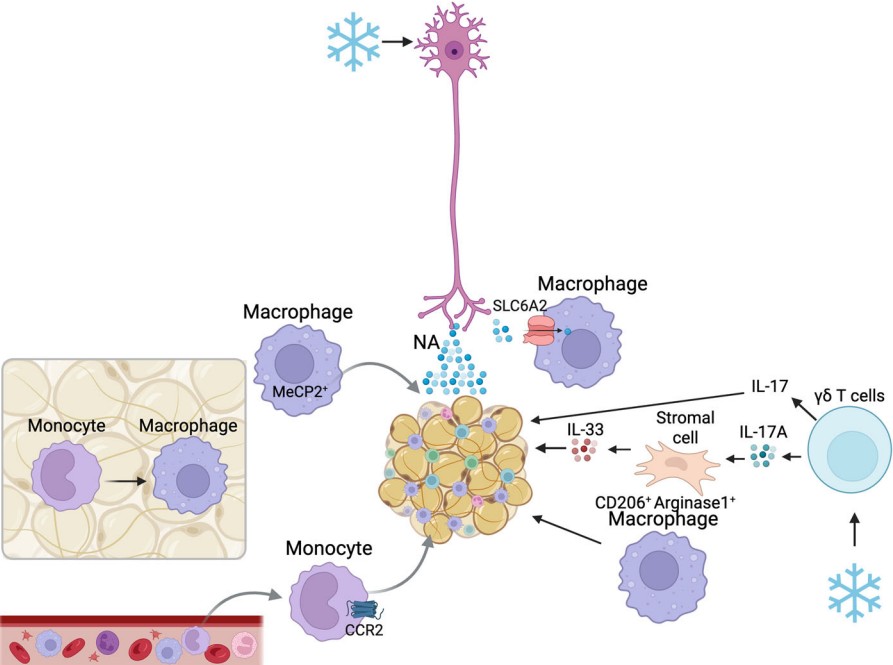

**Figure 3. Immune cells regulate brown adipose tissue (BAT) activity**
Noradrenalin (NA) from sympathetic neurons induces BAT activity in response to cold challenge. BAT macrophages and monocyte populations are diverse with constant replenishment from bone marrow-derived monocytes. Monocytes are required for BAT expansion and lipid accumulation. MeCP2+ macrophages are required to maintain proper BAT innervation. Macrophages express transporters for NA uptake and affect the sympathetic tone of the tissue. Cold challenge induces accumulation of alternative M2 macrophages in BAT to assist thermogenic adaptation. $\gamma\delta$ T cells are required for intact BAT activity and cytokine-induced expression of thermogenic genes after cold exposure. CCR2, C-C chemokine type 2 receptor; MeCP2, methyl-CpG binding protein 2; NA, noradrenaline; SLC6A2, solute carrier family 6 member 2, the norepinephrine transporter; TGF-$\beta$, transforming growth factor $\beta$.

the WAT of mice. This protects mice from diet-induced obesity and improves insulin sensitivity (Rajbhandari et al., 2018; Rajbhandari et al., 2019).

Another subpopulation of adipose tissue macrophages that emerges under obese conditions in mice is known as metabolically activated macrophages and characterized by the expression of the TREM2 protein and a gene signature associated with lipid metabolism (Cottam et al., 2022; Hildreth et al., 2021; Sarvari et al., 2021). Global ablation of the *Trem2* gene prevents the accumulation of lipid-associated macrophages and the acquisition of this specific gene signature. Lipid-associated macrophages form crown-like structures around dead or damaged adipocytes in mice and appear to have a protective effect as they recruit adipocyte-precursor cells that differentiate into beige adipocytes (Cipolletta et al., 2012). Even after weight loss the size of the expanded lipid-associated macrophage subpopulation does not return to normal levels (Cottam et al., 2022). Their absence is associated with adipocyte hypertrophy and accelerated weight gain in mice (Jaitin et al., 2019; Sarvari et al., 2021).

*Role of type 2 immune responses in adipose tissue thermogenic activity.* In addition to macrophages, innate immune cells are enriched in adipose tissue. Eosinophils are an important innate cell population in adipose tissue because of their role in type 2 immune responses that enhance beige and brown adipocyte function and systemic energy homeostasis (Rajbhandari et al., 2018; Rajbhandari et al., 2019). Upon cold exposure, beige adipose tissue releases the protein Meteorin-like (METRNL), which induces thermogenic genes, promotes browning and increases the number of eosinophils. BAT acts in a similar way and releases FGF21, which stimulates chemokine-mediated eosinophil recruitment. Eosinophils produce IL-4 and IL-13, which result in an increase in M2 macrophages, in the differentiation of adipocyte-precursor cells into beige adipocytes and in browning of WAT (Qiu et al., 2014). The abrogation of such responses in mice that do not express the IL-4 receptor or lack downstream STAT6 signalling demonstrates a causal role of type 2 immune responses in BAT thermogenic activity (Nguyen et al., 2011).

Eosinophil and M2 macrophage infiltration, type 2 cytokine signalling and WAT browning are observed when mice are placed on a caloric-restriction diet. These responses are not observed in mice that lack type 2 signalling due to the absence of the IL-4 receptor or STAT6 signalling (Fabbiano et al., 2016), demonstrating the relevance of type 2 immune responses for metabolic health. This was further confirmed by the demonstration that eosinophil-deficient mice have decreased energy expenditure and develop impaired glucose tolerance, when challenged with a high-fat diet (Molofsky et al., 2013). Obesity results in an overall decline in the numbers of eosinophils and ILC2s in visceral WAT in mice (Wu et al., 2011) and subcutaneous WAT in humans (Brestoff et al., 2015).

ILC2s are a second immune cell population that is important in type 2 immune responses. ILC2s require IL-33 for their maintenance, which is produced by stromal cells in response to gamma delta ($\gamma\delta$) T cells. Activation of ILC2s by IL-33 results in beiging of subcutaneous WAT and increased energy expenditure (Brestoff et al., 2015). Beige adipocytes, in turn, contribute to the production of IL-33, thus creating a positive feedback loop. Activation of ILC2s can also occur in response to sympathetic signalling. In this case signals from sympathetic nerves activate $\beta$-adrenergic receptors on mesenchymal cells, which then release the glial cell-derived neurotrophic factor (GDNF) to activate ILC2s (Cardoso et al., 2021).

ILC2s induce beiging via secretion of the opioid-like methionine-enkephalin (MetEnk) peptide, which stimulates opioid receptors on adipocytes (Brestoff et al., 2015), and via IL-5-mediated activation of eosinophils. IL-4 production by eosinophils and IL-33 production by beige adipocytes promote beige adipogenesis via differentiation of beige progenitor cells (Brestoff et al., 2015; Lee et al., 2015). ILC2s also express the costimulatory molecule OX40L and ICOSL and stimulate the proliferation and function of OX40- and ICOS-expressing T-helper 2 (Th$_2$) cells and regulatory CD4$^+$ T cells (T$_{regs}$) (Halim et al., 2018), which contribute further to adipose tissue homeostasis as described later in this review.

In addition to ILC2s, $\gamma\delta$ T cells stimulate the thermogenic activity of adipose tissues. Kohlgruber *et al.* originally described that $\gamma\delta$ T cells produce TNF-$\alpha$ and IL-17A after cold exposure, which stimulates stromal cells to produce IL-33, resulting in the activation of ILC2s and accumulation of T$_{regs}$ in visceral WAT (Kohlgruber et al., 2018). Accordingly, lack of either $\gamma\delta$ T cells or IL-17-receptor expression on adipocytes impairs the induction of thermogenic genes in BAT and inguinal WAT (iWAT), and results in intolerance to cold (Kohlgruber et al., 2018). Hu et al. later established that $\gamma\delta$ T cells produce IL-17F, which induces TGF-$\beta$ in brown adipocytes, thereby promoting sympathetic innervation of BAT (Hu et al., 2020). The blockade of TGF-$\beta$ has the same negative effects on thermogenic gene expression and cold tolerance as the absence of $\gamma\delta$ T cells (Hu et al., 2020). The absence of $\gamma\delta$ T cells also results in reduced expression of the $\beta$3-adrenergic receptor and of genes that are associated with lipolysis in iWAT (Kohlgruber et al., 2018).

Finally, invariant natural killer T (iNKT) cells have emerged as an important immune population that regulates the function of BAT and thereby systemic metabolism (Han et al., 2023; Lynch et al., 2009; Lynch et al., 2012). Like $\gamma\delta$T cells iNKT cells constitute a

tissue-resident immune cell population in adipose tissue (Huh et al., 2013). iNKT cells respond to lipid antigens presented by CD1d on adipocytes. Injecting mice with $\alpha$GalCer, an iNKT cell ligand that is presented in a CD1d-dependent manner, stimulates adipose tissue iNKT cells and promotes beiging of WAT (Lynch et al., 2016).

Adipose tissue iNKT cells can be separated into two subsets, NK1.1 negative and NK1.1 positive, with differential cytokine production. NK1.1-negative iNKT cells express a broad panel of type 2 cytokines (IL-2, IL-4, IL-10 and IL-13), which contribute to the reduction in inflammation and improvement in glycaemic parameters in obese mice (LaMarche et al., 2020). IL-2 secretion supports the proliferation of adipose tissue $T_{reg}$ cells (Lynch et al., 2015). IL-10 secretion is driven by the transcription factor E4BP4, which is upregulated specifically in NKT cells in adipose tissue compared to those in other tissues (Aguiar et al., 2023). IL-10 further promotes the expansion and function of $T_{regs}$ and induces an M2 phenotype in macrophages, thereby preserving adipose tissue homeostasis (Lynch et al., 2015). Consistent with this notion adoptive transfer of NK1.1-negative iNKT cells from adipose tissue of lean mice into obese mice restores decreased metabolic functions in obese mice (LaMarche et al., 2020). NK1.1-positive iNKT cells release interferon-$\gamma$ (IFN-$\gamma$), which stimulates K cell-mediated killing of inflammatory macrophages. This limits the pathogenic expansion of inflammatory macrophages, thereby further preserving adipose tissue homeostasis (Lynch et al., 2015).

*Immune cells and adipose tissue inflammation.* Much of the presented data suggest that adipose tissue homeostasis can be preserved by preventing tissue inflammation. This section highlights several additional immune cell populations that are relevant in this context.

WAT maintains a distinct, clonally expanded population of $T_{regs}$ that differ from their counterparts in lymphoid organs in their transcriptome (driven by the transcription factor PPAR$\gamma$), and their growth- and survival factor dependencies, including dependency on IL-33, as recently reviewed by Becker et al. (2024). In BAT, $T_{regs}$ can be found in the parenchyma, along large neurons and within mesothelial borders. Cold exposure and $\beta$3-adrenergic receptor stimulation increase $T_{reg}$ differentiation, and adoptive transfer of $T_{regs}$ induces thermogenic gene expression in BAT and lipolysis in WAT (Kalin et al., 2017). Conversely, depletion of $CD25^+Foxp3^+$ $T_{regs}$ with either anti-CD25 antibodies or (in Foxp3DTR mice) diphtheria toxin is associated with increased production of inflammatory cytokines by T and NKT cells in BAT and results in mitochondrial dysfunction and an impaired thermogenic response (Medrikova et al., 2015; Zammit et al., 2025).

In addition to immune cell populations that promote adipose tissue homeostasis and function, there are others that appear to have adverse function. $Fc\varepsilon R1^+CD117^+$ mast cells, for example, are found next to microvessels in BAT in mice and are associated with decreased thermogenic BAT activity. Accordingly, genetic deficiency of mast cells results in increased expression of thermogenic genes and energy expenditure. This beneficial effect is maintained when mice are on a high-fat diet challenge, where it is associated with improved glucose homeostasis and reduced weight gain (Liu et al., 2009). Similarly, functional inactivation of mast cells increases thermogenesis and energy expenditure (Zhang et al., 2019).

Several additional immune cell populations have been implicated in the promotion of adipose tissue inflammation in obesity. These include group 1 and group 3 innate lymphoid cells (ILC1s and ILC3s, respectively), which contribute to the accumulation of proinflammatory macrophages in adipose tissue (O'Sullivan et al., 2016). They also secrete proinflammatory cytokines such as TNF-$\alpha$ and IL-1$\beta$, suppress the induction of thermogenic genes in BAT and WAT (Goto et al., 2016) and promote insulin resistance.

As a result of tissue inflammation, iNKT cell numbers decrease in both adipose tissue and circulation of obese humans and mice (Lynch et al., 2009; Lynch et al., 2012). This is likely due to inflammation-induced apoptosis of iNKT cells and mirrors the decrease in iNKT cell numbers in other inflamed tissues, such as the liver (Bolte & Rehermann, 2018). If iNKT cell numbers are merely decreased, they can still be activated in mice via injection of the synthetic glycolipid $\alpha$-galactosylceramide, and this induces browning of WAT and increases energy expenditure and weight loss (Lynch et al., 2012; Lynch et al., 2016). If iNKT cells are completely absent, insulin resistance, enlargement of adipocytes and fat pads and increased weight gain ensue (Lynch et al., 2012; Lynch et al., 2015; Schipper et al., 2012).

Adaptive immune cells also accumulate in adipose tissue of obese mice (Hagglof et al., 2022). In this scenario, T cells express senescence markers and an increased capacity to produce IFN-$\gamma$, which in turn promotes inflammation and inhibits preadipocyte-to-adipocyte differentiation (Dahlquist & Camell, 2022). B cells from adipose tissue of overweight and obese patients express greater levels of the transcription factor *T-bet* and markers of B-cell activation compared to those of lean patients and promote inflammation and metabolic dysregulation. In mice B cells accumulate in obese adipose tissue and produce chemokines that recruit monocytes, inflammatory cytokines that modulate T-cell responses and IgG antibodies that form immune complexes and activate macrophages (Winer et al., 2011).

**Evidence for a microbiota–immune cell–adipose tissue nexus.** Microbiota form a complex ecosystem that has co-evolved with mammals over millions of years and contributes both to their health and to their response to diseases. Commensals are present at all mucosal surfaces and are comprised of many kingdoms, including bacteria, viruses, archaea, protozoa and fungi. The bacterial microbiota in the gut are the most extensively studied kingdom. Their differential effects include absorption and utilization of nutrients; modulation of gut hormone secretion; activation of pathogen-associated pattern receptors; and production of bioactive molecules and metabolites such as short-chain fatty acids, endocannabinoids, amino acid derivates and bile acids, which directly affect adipocytes, adipose tissue homeostasis and host metabolism (reviewed in Cani & Van Hul, 2024; Van Hul & Cani, 2023).

Microbiota and commensals also play a key role in the development and maturation of the host immune system and provide constant tuning of immune responses in lymphoid and non-lymphoid tissues throughout life (Ansaldo et al., 2021; Jordan & Clarke, 2024). Considering the role of the microbiota in shaping the immune response, and the role of the immune system in adipocyte homeostasis as shown in the previous sections, the question arises as to what extent microbiota regulate adipose tissue function and systemic metabolism via the immune response. In this section we will discuss the current evidence for such a microbiota–immune cell–adipose tissue nexus.

*Microbiota affect haematopoiesis and the size of tissue-resident immune cell populations.* Recent studies established that microbiota modulate haematopoietic stem and progenitor cell homeostasis and haematopoiesis. Severe alteration of microbial communities by a 2-week treatment of mice with broad-spectrum antibiotics reduced the numbers of haematopoietic progenitor cells. Although there was no concomitant effect on the number of myeloid progenitor cells, there was a decrease in myelopoiesis with a reduction in granulocytes to the same level as in germ-free mice (Josefsdottir et al., 2017). This is consistent with the notion that myelopoiesis of germ-free mice can be restored by recolonization with gut microbiota (Khosravi et al., 2014). However, because research mice in conventional barrier facilities lack many commensals of their free-living counterparts (Rosshart et al., 2019), it is not surprising that they still have lower numbers of immune cells in lymphoid and non-lymphoid organs than mice on the same genetic background that are colonized with microbiota and commensals from wild mice (Oh et al., 2025). The latter have increased numbers of myeloid cells, in particular eosinophils, neutrophils and monocytes (Oh et al., 2025), which – as described in the previous section – are relevant for adipose tissue homeo-

stasis. Similar patterns of expanded myeloid and lymphoid immune cell populations are observed in research mice that acquired an increased diversity of microbes and pathogens by cohousing with outbred 'dirty' pet-store mice (Beura et al., 2016; Rehermann et al., 2025).

Of relevance to this review, inbred research mice are resistant to diet-induced obesity when colonized with complex microbiota and commensals of wild mice. This is associated with increased glucose uptake in BAT and increased body temperature, suggesting increased thermogenesis (Hild et al., 2021). This protection depends on colonization with natural microbiota during a short window of time in the first 2 weeks of life and cannot be induced when colonization occurs at later stages of life (Hild et al., 2021). Consistent with these results, germ-free mice colonized with the gut microbiota of wild non-human primates gain less weight when fed with a low- or high-fibre diet than germ-free mice that received the gut microbiota of captive non-human primates (Sidiropoulos et al., 2020). These data suggest that the natural, complex microbiota and likely the immune responses that they induce have co-evolved with mammals to confer metabolic health benefits.

*Microbiota and commensals affect type 2 immune responses and metabolic health.* Several studies have shown that gut microbiota regulate and control type 2 immune responses. Germ-free mice and mice that are depleted of microbiota by antibiotic treatment exhibit increased levels of eosinophils, M2 macrophages and type 2 cytokines in iWAT (Suarez-Zamorano et al., 2015). This type 2 immune response in microbiota-depleted mice was associated with increased browning of WAT, reduction in fat mass and improvement in glucose tolerance and insulin sensitivity, without any effect on BAT activity. Later studies by Li and colleagues reported an impaired thermogenic response of BAT in microbiota-depleted mice and reduced UCP-1 expression and browning of WAT (Li et al., 2019). Similar to germ-free mice (Backhed et al., 2004), these microbiota-depleted mice exhibited increased caecal size and intestinal content (Li et al., 2019). This may have affected nutrient absorption and reduction in metabolites such as short-chain fatty acids that are typically produced by microbial fermentation of dietary fibre. Indeed UCP1 expression and thermogenesis were rescued when the short-chain fatty acid butyrate was administered to antibiotic-treated mice (Li et al., 2019). Thus, intact microbiota appear to be an essential component of the thermoregulatory response.

Commensals such as helminths that are mostly eliminated in industrialized populations due to improved hygiene and medication but have long been part of the normal gut flora do also modulate adipose tissue homeostasis by increasing type 2 immune responses (Loke & Lim, 2015). Indeed experimental infection of

obese mice with the migratory helminth *Nippostrongylus brasiliensis* activates ILC2s and increases the eosinophil population in the adipose tissue of the mice, thereby contributing to a reduction in WAT mass and to an improvement in metabolic parameters (Molofsky et al., 2013; Wu et al., 2011). Along the same line treatment of mice with the *Schistosoma mansoni* helminth egg-related antigen T2 RNase $\omega$1 induces the expression of UCP1 in mitochondria of brown and beige adipocytes, thereby regulating thermogenesis and energy balance in BAT and WAT and browning of WAT. Transfer of M2 macrophages from such mice to other mice likewise induces browning of WAT (Hams et al., 2016; Su et al., 2018).

*Microbiota affect adipose tissue inflammation.* It is well established that diet and obesity alter the gut microbiota composition with a major effect on the Firmicutes-to-Bacteroidetes ratio (Ley et al., 2005). Transfer of such altered microbiota from obese mice or humans into germ-free mice reproduces the obese phenotype (Ridaura et al., 2013; Turnbaugh et al., 2006). This is associated with increased gut permeability and translocation of bacteria, their DNA and lipopolysaccharides (LPS) and endotoxins into the bloodstream and adipose tissue (Cani & Van Hul, 2024; Cani et al., 2007, 2008; Cani, 2018; DiMattia et al., 2024; Van Hul & Cani, 2023). LPS binds to the toll-like receptor 4 (TLR4)/CD14 complex on preadipocytes, adipocytes and macrophages. This promotes the secretion of proinflammatory cytokines (Cani & Van Hul, 2024), which interfere with adipogenesis, and downregulates UCP1 expression and mitochondrial respiration (Bae et al., 2014; Okla et al., 2015). Accordingly, mice that do not express TLR4 on bone marrow cells have reduced adipose tissue inflammation and improved metabolic parameters even if they are obese (Saberi et al., 2009). Similarly inflammation can be reduced via a reduction in LPS biosynthesis, which occurs when gut microbiota are altered via every-other-day fasting or depleted (Li et al., 2017). These examples illustrate that gut microbiota and related immune responses are affected by the metabolic state of the host and can transfer this metabolic phenotype to other hosts.

The effects of obesity- and diet-induced inflammation extend beyond the adipose tissue to the hypothalamus. In fact inflammation in the hypothalamus is an early cause of leptin resistance, which promotes overeating and obesity. Glucagon-like peptide-1 (GLP-1), which is produced by L-cells in the colon in response to short-chain fatty acids (Tolhurst et al., 2012), has a protective effect, which depends on GLP-1 receptor expression and signalling in astrocytes (Heiss et al., 2021). Immune cells in adipose tissue also express the GLP-1 receptor, and GLP-1 has been shown to activate macrophages and induce their M2 polarization (Shiraishi et al., 2012). Similarly the GLP-1 receptor agonist, liraglutide, has been shown to activate iNKT cells, which in turn promote FGF21-induced activation of BAT and browning of WAT. The resulting increased thermogenesis improved glycaemic and weight control in both humans and mice (Lynch et al., 2016). Because microbiota-mediated digestion of fibres results in the production of short-chain fatty acids, which stimulate GLP-1 production by L-cells in the colon, altered gut microbiota and diet may affect type 2 immune responses and adipose tissue homeostasis in this manner. Similarly it is possible that therapeutic modulation of the gut microbiota can be applied to induce protective effects in adipose tissue.

## Conclusion and further directions

WAT and BAT constitute a major metabolic organ that plays a crucial role in regulating body energy homeostasis beyond just accumulating and expending energy. Cross-talk with tissue-resident and recruited immune cells contributes not only to the maintenance of tissue homeostasis and function in health but also to inflammatory responses that inhibit adipose tissue function in obesity. In recent years the gut microbiota emerged as a factor that modulates adipose tissue homeostasis and function and thereby affects systemic metabolism. Many mechanisms, by which it regulates adipose tissue function, were elucidated, including but not limited to metabolites, endocannabinoids, LPS and bile acids (Cani & Van Hul, 2024; Van Hul & Cani, 2023).

Considering the role of microbiota in shaping and educating the immune system on one hand (Ansaldo et al., 2021; Jordan & Clarke, 2024), and the role of the immune system in regulating adipose tissue homeostasis on the other (Trim & Lynch, 2022), it is intuitive to postulate that the gut microbiota also regulate adipose tissue function by fine-tuning immune cell composition and function in adipose tissue. Although some studies insinuate the existence of such a microbiota–immune cell–adipose tissue nexus, limited studies provide direct evidence of the underlying pathways. The following section identifies areas for future research and the associated challenges.

**Early life.** Future studies should also include an analysis of the effects of microbiota and immune responses on the development of brown and beige adipose tissue in early life, when all three undergo rapid developmental adjustments and may cross-regulate each other. Studies in humans have identified interfering factors, such as early treatment with antibiotics that results in an increased risk of obesity later in life and described effects that accumulate over generations (Sonnenburg et al., 2016). Future studies may aim to identify protective factors and mechanisms that result in enhanced BAT and beige adipose tissue

formation and long-term metabolic health. Such an association was shown in mice and humans between preconception cold exposure of males and increased innervation, vascularization and ultimately BAT activity and metabolism of the offspring (Sun et al., 2018). It would be interesting to investigate whether microbiota-mediated induction and fine-tuning of type 2 immune responses, which also occur in a critical window of time in early life (McCoy et al., 2018), have similar long-term effects.

**Studies involving the complete meta-organism.** Microbiome research should be extended to kingdoms other than bacteria, include commensals and extend from the gut to other organs. These studies may, for example, include the skin, the largest organ of the body, which is in close proximity to BAT, and to signals from microbe-associated products that are directly present in the adipose tissue. Such research may ultimately identify druggable pathways and methods of intervention in early life that protect against adipose tissue dysfunction and metabolic abnormalities.

**Translation to humans.** Adult humans have a much lower amount of BAT than mice, and obese persons tend to have even less activated BAT. However [18F]fluorodeoxyglucose (FDG)-PET/CT scans have demonstrated active BAT in humans (Cypess et al., 2009; Cypess, 2023), and there is now solid evidence that cold-induced BAT activation in humans enhances clearance of glucose fatty acids and triglycerides from the circulation (Orava et al., 2011; Blondin et al., 2014). Indeed BAT activity is associated with metabolic effects, including cardiometabolic health in humans (Becher et al., 2021; Herz et al., 2022). Combined, the current data suggest that adipose tissue can be considered a tissue of interest to promote metabolic health. The identification of mechanisms that modulate BAT or beige adipose tissue activity may provide additional therapeutic targets to combat metabolic disease.

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

## Additional information

### Competing interests

The authors declare no conflicts of interest.

## Author contributions

S.A. and B.R. conceptualized and wrote the manuscript. S.A. created the figures using BioRender.com. All authors approved the final version of the manuscript and qualify for authorship. All those who qualify for authorship are listed.

## Funding

This research was supported by the Intramural Research Program of the National Institute of Diabetes and Digestive and Kidney Diseases (NIDDK) within the National Institutes of Health (NIH). The contributions of the NIH authors were made as part of their official duties as NIH federal employees, are in compliance with agency policy requirements and are considered Works of the United States Government. However the findings and conclusions presented in this paper are those of the authors and do not necessarily reflect the views of the NIH or the U.S. Department of Health and Human Services.

## Keywords

adipose tissue, cytokine, immune response, microbiome

## Supporting information

Additional supporting information can be found online in the Supporting Information section at the end of the HTML view of the article. Supporting information files available:

**Peer Review History**

