## [Peer Review History · The Journal of Physiology]

21 August 2025

Dear Dr Rehermann,

Re: JP-TR-2025-286422 **"The Role of Immune Responses and Microbiota in Adipose Tissue Homeostasis"** by Barbara Rehermann and Shahar Azar

Thank you for submitting your manuscript to The Journal of Physiology. It has been assessed by a Reviewing Editor and by 3 expert referees and we are pleased to tell you that it is acceptable for publication following satisfactory revision.

Your revised manuscript should be submitted online using the link in your Author Tasks: <https://jp.msubmit.net/cgi-bin/main.plex>

ABSTRACT FIGURES: Authors may use The Journal's premium BioRender account to create/redraw their Abstract Figures (and any other suitable schematic figure). Information on how to access this account is here: <https://physoc.onlinelibrary.wiley.com/journal/14697793/biorender-access>.

REVISION CHECKLIST: Upload a full Response to Referees file. To create your 'Response to Referees' copy all the reports, including any comments from the Senior and Reviewing Editors, into a Microsoft Word, or similar, file and respond to each point, using font or background colour to distinguish comments and responses and upload as the required file type.

- 'Potential Cover Art' for consideration as the issue's cover image
- Appropriate Supporting Information (Video, audio or data set: see <https://jp.msubmit.net/cgi-bin/main.plex>)

We look forward to receiving your revised submission.

Yours sincerely,

Kim Barrett
Senior Editor
The Journal of Physiology

REQUIRED ITEMS

- Your MS must include a complete "Additional information section" with the following 4 headings and content:

Competing Interests: A statement regarding competing interests. If there are no competing interests, a statement to this effect must be included. All authors should disclose any conflict of interest in accordance with journal policy.

Author contributions: Each author should take responsibility for a particular section of the study and have contributed to writing the paper. Acquisition of funding, administrative support or the collection of data alone does not justify authorship; these contributions to the study should be listed in the Acknowledgements. Additional information such as 'X and Y have contributed equally to this work' may be added as a footnote on the title page.

It must be stated that all authors approved the final version of the manuscript and that all persons designated as authors qualify for authorship, and all those who qualify for authorship are listed.

Funding: Authors must indicate all sources of funding, including grant numbers. If authors have not received funding, this must be stated.

It is the responsibility of authors funded by RCUK to adhere to their policy regarding funding sources and underlying research material. The policy requires funding information to be included within the acknowledgement section of a paper. Guidance on how to acknowledge funding information is provided by the Research Information Network. The policy also requires all research papers, if applicable, to include a statement on how any underlying research materials, such as data, samples or models, can be accessed. However, the policy does not require that the data must be made open. If there are considered to be good or compelling reasons to protect access to the data, for example commercial confidentiality or legitimate sensitivities around data derived from potentially identifiable human participants, these should be included in the statement.

Acknowledgements: Acknowledgements should be the minimum consistent with courtesy. The wording of acknowledgements of scientific assistance or advice must have been seen and approved by the persons concerned. This section should not include details of funding.

EDITOR COMMENTS

Reviewing Editor:

Thank you for your submission of this interesting review for the upcoming special issue of the Journal of Physiology on "The role of the gut in obesity - new insights into the physiology of therapeutic approaches". Three experts in the field have reviewed the manuscript and while they generally found the review to be well-written and informative, there were some significant concerns, particularly from Reviewer 2 who felt that there was a lack of novelty in the sections relating to immune regulation of adipose tissue, while sections relating to the microbiota were not sufficiently developed. Reviewer 1 also felt that the graphical abstract should be reconsidered as it does not adequately summarise the information discussed in the manuscript. I agree that addressing each the Reviewers comments will enhance the impact of this timely review.

Please also see 'Required Items' above.

REFEREE COMMENTS

Referee #1:

In this review Rehermann and Azar give an impressive overview on adipose tissue homeostasis and immune cells response and the influence of the gut microbiota on both of them. The manuscript is well written, up-to-date and covers several important aspects of this intertwined relation. Just a few minor aspects to improve the readability and the quality of this piece of work:

- The layout of the paragraph "Immune cell landscape of lean and obese adipose tissue" should be thought over. There are several players in bold, but not always coherently, therefore a more precise approach, maybe with separate subparagraph, would help the reader easing the flow.

- Also, the graphical abstract should be reconsidered. As it is, it does not provide any info, while it should be more clear how the different players affect BAT and WAT homeostasis.

- Since there is the last paragraph titled "Can gut microbiota be modulated to affect adipose tissue homeostasis?"(which I would put in bold) I would probably add a table about current modulation of the GM via probiotic administration in humans that could have an effect on obesity and/or metabolic diseases.

Referee #2:

JP-TR-2025-286422

Azar and Rehermann

Summary and Critiques

Based on the paper title and abstract, this reviewer was initially enthusiastic since the topic area of the gut microbiota-immune system-adipose tissue nexus is an emerging area. Yet, the main body of the work deals with a review and primer on immune cell-adipose interactions. To the authors' credit, this was covered in significant depth, but the topic has already been addressed in many reviews in the extant literature. This limits the novelty and impact of the review. In addition, the section that does address the microbiome-immune-adipose nexus (along with the related Graphical Abstract) is largely speculative considering the limited data in this arena and lack of deeper knowledge regarding specific microbiota signals involved.

Comments:

(1) A general comment is that the paper leans on dogma and would benefit from also citing null results and counter-results in some places. Examples: The oft-cited but implied role of beige UCP1 to promote thermogenesis even though the induction of the protein is typically very, very low in absolute terms; The proposed exercise increase in 12,13-diHOME and related effects, even though this finding has often not been seen by other investigators; Adiponectin secretion by myocytes is (like the UCP1 example) very, very low; Adipocyte hyperplasia is limited under normal conditions relative to hypertrophy; is it truly "well established" that obesity drives major changes in the gut microbiome, especially in humans?

(2) Another general comment is that while the authors do a nice job here-and-there to weave into the narrative the species and experimental details, often this is missing and the text reads like everything is fact no matter what the model. Of course, many principles in mouse models recapitulate in humans but many don't, and this is increasingly appreciated in the literature on the microbiome. The paper would be improved by weaving in model, doses, conditions, etc. as appropriate to help back up interpretive assertions. Example: L. 335 mentions administration of IL-4 increasing energy expenditure, yet was this a pharmacological amount? (and there are other examples like this)

(3) In the section related to gut microbiome and adipose, it is not immediately clear why neuroinflammation or GLP-1 biology are directly relevant to the subject matter under discussion.

(4) The authors pay little to no attention to the key role of adipose macrophages in extracellular matrix remodeling and architecture, yet this is a central phenomenon that enables normal adipose expansion and dysfunctional ECM is a player contributing to inflammation in some cases of obesity.

(5) Minor issues:

a) "adipokine" is a commonly used colloquial term but not scientifically correct since "kine" refers to movement and adipocytes don't move in response to the adipose-derived hormones.

b) Some text in the figures are very tiny and hard to read, especially Figure 2

c) This reviewer challenges the definitions and completeness of the Figure 1 and related discussion of adipose depots. For instance, the perirenal fat pad is distinct from the retroperitoneal fat pad despite both pads being closely juxtaposed; the RP

pad is not mentioned for mice in the figure or text. In addition, perigonadal pads are sometimes included in the VAT category by some authors but this is difficult to defend based on physiology and pathophysiology related to VAT in humans (i.e., that links to mesenteric fat most commonly); perigonadal is likely its own distinct category. There is also a subcutaneous pad under the skin around the ribcage of mice and that can become more substantial with obesity; it is not mentioned in the figure or text.

d) L. 247, the "free fatty acids" are not likely from "excess dietary lipids" but rather they are from adipose insulin resistance and concomitant maintenance of higher lipolysis.

e) L. 413, define "BAT activity"

f) Several places in the text and also in Figure 2: do the authors mean 'beta-3' and not 'beta-2' ?

Referee #3:

JR - TP 2025 286422

The Role of Immune Responses and Microbiota in Adipose Tissue Homeostasis

Commissioned Review

Thorough description of the distribution and function of white, brown and beige adipocytes in humans and rodents. Good use of supporting figures.

The Introduction / contextualisation could be further enhanced and supported with appropriate citations. State goal of the review.

Justify why is it important / timely to review studies in this field?

To my mind, the novel and most interesting aspect of this review relates to how the microbiome, via immune modulation, impacts on adipose tissue homeostasis. This could be conveyed more clearly. Indeed, in this context, I would suggest reworking the review to introduce the effects of microbiota on adipose tissue first, and then relating that to the changes in immune cell function.

Provide additional context as to how SCFAs modulate GLP-1 secretion (L-cells).

The translation to humans section on page 18 does not directly refer back to the work in rodents - doesn't link microbial products to changes in adipose tissue.

Add concrete conclusions - does the literature support the original hypothesis? Remaining questions have been clearly described.

END OF COMMENTS

Response to Editor and Reviewer Comments

REQUIRED ITEMS

- Your MS must include a complete "Additional information section" with the following 4 headings and content:

Competing Interests: A statement regarding competing interests. If there are no competing interests, a statement to this effect must be included. All authors should disclose any conflict of interest in accordance with journal policy.

Author contributions: Each author should take responsibility for a particular section of the study and have contributed to writing the paper. Acquisition of funding, administrative support or the collection of data alone does not justify authorship; these contributions to the study should be listed in the Acknowledgements. Additional information such as 'X and Y have contributed equally to this work' may be added as a footnote on the title page.

It must be stated that all authors approved the final version of the manuscript and that all persons designated as authors qualify for authorship, and all those who qualify for authorship are listed.

Funding: Authors must indicate all sources of funding, including grant numbers. If authors have not received funding, this must be stated.

Author response:

We have added the 'Additional information section' with the four subsections.

EDITOR COMMENTS

Reviewing Editor:

Thank you for your submission of this interesting review for the upcoming special issue of the Journal of Physiology on "The role of the gut in obesity - new insights into the physiology of therapeutic approaches". Three experts in the field have reviewed the manuscript and while they generally found the review to be well-written and informative, there were some significant concerns, particularly from Reviewer 2 who felt that there was a lack of novelty in the sections relating to immune regulation of adipose tissue, while sections relating to the microbiota were not sufficiently developed. Reviewer 1 also felt that the graphical abstract should be reconsidered as it does not adequately summarise the information discussed in the manuscript. I agree that addressing each the Reviewers comments will enhance the impact of this timely review.

Please also see 'Required Items' above.

Author response:

We have added the 'Additional information section' with the four subsections.

REFeree COMMENTS

Referee #1:

In this review Rehermann and Azar give an impressive overview on adipose tissue homeostasis and immune cells response and the influence of the gut microbiota on both of them. The manuscript is well written, up-to-date and covers several important aspects of this intertwined relation. Just a few minor aspects to improve the readability and the quality of this piece of work:

Author response:

Thank you for the positive comments, it was a massive effort for us.

- The layout of the paragraph "Immune cell landscape of lean and obese adipose tissue" should be thought over. There are several players in bold, but not always coherently, therefore a more precise approach, maybe with separate subparagraph, would help the reader easing the flow.

Author response:

We have revised this section and added subparagraphs.

- Also, the graphical abstract should be reconsidered. As it is, it does not provide any info, while it should be more clear how the different players affect BAT and WAT homeostasis.

Author response:

We revised the graphical abstract to better reflect the concept of this review, as stated in the figure legends. It is not possible to add detail on the many mechanisms that affect adipose tissue homeostasis. We provided these details in the individual figures of the manuscript.

- Since there is the last paragraph titled "Can gut microbiota be modulated to affect adipose tissue homeostasis?"(which I would put in bold) I would probably add a table about current modulation of the GM via probiotic administration in humans that could have an effect on obesity and/or metabolic diseases.

Author response:

We have restructured this part of the manuscript according to comments from reviewer 2 and changed the paragraph title to better reflect what it is covered. Our aim was to review the physiology of adipose tissue with a focus on the emerging microbiome – immune response – adipose tissue axis. We feel that it is beyond the scope of this manuscript to cover the field of probiotics.

Referee #2:

**JP-TR-2025-286422, Azar and Rehermann,
Summary and Critiques**

Based on the paper title and abstract, this reviewer was initially enthusiastic since the topic area of the gut microbiota-immune system-adipose tissue nexus is an emerging area. Yet, the main body of the work deals with a review and primer on immune cell-adipose interactions. To the authors' credit, this was covered in significant depth, but the topic has already been addressed in many reviews in the extant literature. This limits the novelty and impact of the review. In addition, the section that does address the microbiome-immune-adipose nexus (along with the related Graphical Abstract) is largely speculative considering the limited data in this arena and lack of deeper knowledge regarding specific microbiota signals involved.

We have revised the manuscript according to the reviewer's comments, which we appreciate.

Comments:

(1) A general comment is that the paper leans on dogma and would benefit from also citing null results and counter-results in some places. Examples: The oft-cited but implied role of beige UCP1 to promote thermogenesis even though the induction of the protein is typically very, very low in absolute terms; The proposed exercise increase in 12,13-diHOME and related effects, even though this finding has often not been seen by other investigators; Adiponectin secretion by myocytes is (like the UCP1 example) very, very low; Adipocyte hyperplasia is limited under normal conditions relative to hypertrophy; is it truly "well established" that obesity drives major changes in the gut microbiome, especially in humans?

Author response:

Thank you for these comments, we have now revised the manuscript according to the reviewer comments and provided a more differentiated view in other places as well.

(2) Another general comment is that while the authors do a nice job here-and-there to weave into the narrative the species and experimental details, often this is missing and the text reads like everything is fact no matter what the model. Of course, many principles in mouse models recapitulate in humans but many don't, and this is increasingly appreciated in the literature on the microbiome. The paper would be improved by weaving in model, doses, conditions, etc. as appropriate to help back up interpretive assertions. Example: L. 335 mentions administration of IL-4 increasing energy expenditure, yet was this a pharmacological amount? (and there are other examples like this)

Author response:

We have included a comparison of mouse and human where appropriate and deleted the statement on IL-4 increasing energy expenditure and the reference, since any injected recombinant IL-4 exceeds the physiologic level of IL-4 at the tissue site.

(3) In the section related to gut microbiome and adipose, it is not immediately clear why neuroinflammation or GLP-1 biology are directly relevant to the subject matter under discussion.

Author response:

We have revised this section.

(4) The authors pay little to no attention to the key role of adipose macrophages in extracellular matrix remodeling and architecture, yet this is a central phenomenon that enables normal adipose expansion and dysfunctional ECM is a player contributing to inflammation in some cases of obesity.

Author response:

Thank you for pointing this out. We had just included a single sentence and reference (Gallerand et al., 2021) and have now expanded this part and added a review (Sun et al., 2023)

(5) Minor issues:

a) "adipokine" is a commonly used colloquial term but not scientifically correct since "kine" refers to movement and adipocytes don't move in response to the adipose-derived hormones.

Author response:

We have deleted the term "adipokine" throughout the manuscript.

b) Some text in the figures are very tiny and hard to read, especially Figure 2

Author response:

We have enlarged the text in the figures.

c) This reviewer challenges the definitions and completeness of the Figure 1 and related discussion of adipose depots. For instance, the perirenal fat pad is distinct from the retroperitoneal fat pad despite both pads being closely juxtaposed; the RP pad is not mentioned for mice in the figure or text. In addition, perigonadal pads are sometimes included in the VAT category by some authors but this is difficult to defend based on physiology and pathophysiology related to VAT in humans (i.e., that links to mesenteric fat most commonly); perigonadal is likely its own distinct category. There is also a subcutaneous pad under the skin around the ribcage of mice and that can become more substantial with obesity; it is not mentioned in the figure or text.

Author response:

We have revised the figure and the related text according to the reviewer's comments.

d) L. 247, the "free fatty acids" are not likely from "excess dietary lipids" but rather they are from adipose insulin resistance and concomitant maintenance of higher lipolysis.

Author response:

Yes, we agree. We had correctly stated this in the second paragraph of the section 'Adipose tissue and its role in metabolic homeostasis' and have now revised the second statement in the section 'Immune cell landscape of lean and obese adipose tissue'.

e) L. 413, define "BAT activity"

Author response:

We have now defined it as thermogenic activity.

f) Several places in the text and also in Figure 2: do the authors mean 'beta-3' and not 'beta-2' ?

Author response:

We have corrected this. We had referenced Cardoso et al. who had written about beta-2-adrenergic receptors on mesenchymal cells (Nature 2021).

Referee #3:

JR - TP 2025 286422

The Role of Immune Responses and Microbiota in Adipose Tissue Homeostasis

Commissioned Review

Thorough description of the distribution and function of white, brown and beige adipocytes in humans and rodents. Good use of supporting figures.

Author response:

Thank you for the positive comment.

The Introduction / contextualisation could be further enhanced and supported with appropriate citations. State goal of the review.

Author response:

We have revised the introduction and included a statement of the goal of the review.

Justify why is it important / timely to review studies in this field?

Author response:

We have added a justification.

To my mind, the novel and most interesting aspect of this review relates to how the microbiome, via immune modulation, impacts on adipose tissue homeostasis. This could be conveyed more clearly. Indeed, in this context, I would suggest re-working the review to introduce the effects of microbiota on adipose tissue first, and then relating that to the changes in immune cell function.

Author response:

We have revised the review to provide a primer on adipose tissue physiology and function and on the role of the immune responses in brown and white adipose tissue homeostasis versus obesity-induced inflammation. Immune responses are the main part of the review. In the third part of the review we discuss evidence for a microbiota-immune cell-adipose tissue nexus and identify research questions and challenges for future work.

Provide additional context as to how SCFAs modulate GLP-1 secretion (L-cells).

Author response:

We have provided additional context.

The translation to humans section on page 18 does not directly refer back to the work in rodents - doesn't link microbial products to changes in adipose tissue.

Author response:

We have revised this part of the manuscript.

Add concrete conclusions - does the literature support the original hypothesis? Remaining questions have been clearly described.

Author response:

We have revised the conclusions and expanded the section on open questions and future work.

National Institutes of Health
National Institute of Diabetes and
Digestive and Kidney Diseases
Bethesda, Maryland 20892

September 30, 2025

Professor Kim Barrett
Editor-in-Chief
The Journal of Physiology

JP-TR-2024-286422R1 The role of immune responses and microbiota in adipose tissue homeostasis

Dear Professor Barrett:

I am herewith submitting the revised invited topical review 'The role of immune responses and microbiota in adipose tissue homeostasis' by Shahar Azar and Barbara Rehermann including a point-by-point response to the editor and reviewer comments. The figures were produced with an institutional BioRender license, which I include. Thank again for the opportunity to write this review to the J Physiology.

Sincerely,

Barbara Rehermann, MD
Chief, Immunology Section, LDB, NIDDK
National Institutes of Health, DHHS
Bethesda, MD 20892-1800

Phone: +1 (301) 402-7144,
Email: Rehermann@nih.gov

Dear Dr Rehermann,

Re: JP-TR-2025-286422R1 "**The Role of Immune Responses and Microbiota in Adipose Tissue Homeostasis**" by Barbara Rehermann and Shahar Azar

We are pleased to tell you that your paper has been accepted for publication in The Journal of Physiology.

Authors should note that it is too late at this point to offer corrections prior to proofing. Major corrections at proof stage, such as changes to figures, will be referred to the Editors for approval before they can be incorporated. Only minor changes, such as to style and consistency, should be made at proof stage. Changes that need to be made after proof stage will usually require a formal correction notice.

Yours sincerely,

Kim Barrett
Senior Editor
The Journal of Physiology

P.S. - You can help your research get the attention it deserves! Check out Wiley's free Promotion Guide for best-practice recommendations for promoting your work at www.wileyauthors.com/eeo/guide. You can learn more about Wiley Editing Services which offers professional video, design, and writing services to create shareable video abstracts, infographics, conference posters, lay summaries, and research news stories for your research at www.wileyauthors.com/eeo/promotion.

IMPORTANT NOTICE ABOUT OPEN ACCESS: To assist authors whose funding agencies mandate public access to published research findings sooner than 12 months after publication, The Journal of Physiology allows authors to pay an Open Access (OA) fee to have their papers made freely available immediately on publication.

You can check if your funder or institution has a Wiley Open Access Account here: <https://authorservices.wiley.com/author-resources/Journal-Authors/licensing-and-open-access/open-access/author-compliance-tool.html>.

EDITOR COMMENTS

Reviewing Editor:

Dear Dr. Reherman, Thank you for the significant revisions made to this interesting and highly informative manuscript. These revisions have improved the overall flow and impact of the review and I feel it will make a superb addition to the upcoming special issue on the Role of the Gut in Obesity.

REFeree COMMENTS

Referee #1:

The author have answered my comments satisfactorily.

Referee #2:

This reviewer appreciates the effort to add more detail and clarity around specifics noted in the prior critique, thank you. The core critiques of the paper, however, are not changed. Specifically, the bulk of the review focuses on adipose biology for which there are many existing reviews in the literature and many which go far deeper into the various aspects of adipose biology. Most of the coverage here is superficial, with the exception of more information related to adipose and immune system. Second, the links between the microbiome and adipose/immune interactions remain tenuous and speculative.

Referee #3:

The authors have made considerable, favourable amendments to their manuscript in response to the comments from the reviewers.

The re-submitted version of this review has addressed my comments satisfactorily resulting in a more comprehensive and cohesive review relating to microbiota, immune responses and adipose tissue homeostasis.

There remains a few typographical errors in the paper, but otherwise, I have no further comments.